# Reconfigurable photonic generation of broadband chirped waveforms using a single CW laser and low-frequency electronics

Hugues Guillet de Chatellus[1,2], Luis Romero Cortés[2], Côme Schnébelin[1], Maurizio Burla[2,3] & José Azaña[2]

Broadband radio-frequency chirped waveforms (RFCWs) with dynamically tunable parameters are of fundamental interest to many practical applications. Recently, photonic-assisted solutions have been demonstrated to overcome the bandwidth and flexibility constraints of electronic RFCW generation techniques. However, state-of-the-art photonic techniques involve broadband mode-locked lasers, complex dual laser systems, or fast electronics, increasing significantly the complexity and cost of the resulting platforms. Here we demonstrate a novel concept for photonic generation of broadband RFCWs using a simple architecture, involving a single CW laser, a recirculating frequency-shifting loop, and standard low-frequency electronics. All the chirp waveform parameters, namely sign and value of the chirp rate, central frequency and bandwidth, duration and repetition rate, are easily reconfigurable. We report the generation of mutually coherent RF chirps, with bandwidth above 28 GHz, and time-bandwidth product exceeding 1000, limited by the available detection bandwidth. The capabilities of this simple platform fulfill the stringent requirements for real-world applications.

[1] Univ. Grenoble Alpes, CNRS, LIPHY, 38000 Grenoble, France. [2] Institut National de la Recherche Scientifique – Energie, Matériaux et Télécommunications (INRS-EMT), Varennes, QC J3X1S2, Canada. [3] Institute of Electromagnetic Fields, ETH Zurich, Gloriastrasse 35, 8092 Zurich, Switzerland. Correspondence and requests for materials should be addressed to H.G.d C. (email: hugues.guilletdechatellus@univ-grenoble-alpes.fr)

Radio-frequency chirped waveforms (RFCW), or "RF chirps" are broadband waveforms whose instantaneous frequency exhibits well-defined temporal variations. Waveforms with linear frequency modulation, or "linear RF chirps" are of primary importance for applications such as radar systems[1–3], biomedical imaging[4], testing of RF components, and chirped-pulse Fourier transform microwave spectroscopy[5, 6]. Sources of RFCWs for real-world applications are required to provide signals with easily reconfigurable parameters (bandwidth, central frequency, chirp rate/sign, envelope, and time-bandwidth product) (Fig. 1). Moreover a high degree of repeatability and pulse-to-pulse coherence are highly desired, e.g., for sensitivity enhancement by pulse accumulation, and/or for high-resolution coherent radar techniques (pulse compression, frequency-modulated continuous wave, or FMCW radar), among others[2]. In practice, electronic arbitrary waveform generators (AWGs) offer a remarkable performance for the generation of RFCW in terms of ease of use and programmability. However, these devices are bulky and expensive, and the capabilities of state-of-the-art digital-to-analog converters constrain the achievable bandwidth up to a few tens of GHz.

Photonic generation of arbitrary waveforms, including linear RF chirps, present an attractive alternative to purely electronic approaches[7–9]. In fact, the bandwidth of optical waveforms can easily reach the THz range, while optical systems are immune to electromagnetic interference. Moreover, since the advent of fiber-optics technology, optical solutions can be implemented at a reasonable cost, based on available telecommunication components, and integrated solutions are foreseeable[10, 11]. Therefore, over the last 15 years, extensive efforts have been devoted by the scientific community to develop efficient photonic-based sources of linearly chirped RF signals. Numerous approaches have been proposed, including direct frequency modulation[12], space-to-time mapping[13–15] and Fourier transform pulse shaping followed by frequency-to-time mapping (FTM)[16, 17].

The latter technique has become particularly popular due to the availability of pulse shapers and dispersive lines in the telecom C band[18–23]. The chirp rate is set by both the spectral filter and the amount of group-velocity dispersion (GVD), which enables some degree of reconfigurability, although limited by the lack of available tunable dispersive lines. However, all practical implementations based on dispersive pulse shaping reported so far, make use of bulky broadband mode-locked (ML) lasers, resulting in high cost, complexity, limited scalability, and relatively low long-term reliability outside a controlled lab environment. Another popular approach for generating RFCWs is based on the down-conversion of optical chirps to the RF domain: mixing a chirped optical waveform with a single-frequency laser in a fast photodetector enables generation of the desired RFCW[24]. The chirped optical waveform can be produced by dispersive propagation of a ML laser[25–27], incurring in the aforementioned drawbacks. Alternatively, it can be produced by linear frequency modulation of a continuous wave (CW) laser by controlled injection seeding[28] or frequency sweeping[29]. However, solutions based on heterodyne beatings between two different lasers suffer from limited chirp repeatability, and temporal pulse-to-pulse coherence. Recently, some different configurations have been proposed that make use of a single CW laser[30, 31]. An optical chirp is produced by electronic modulation of the laser, and the RF chirp is produced by a self-heterodyning technique. This scheme has several advantages over generation schemes based on ML lasers or two-laser systems, i.e., simplicity and increased signal duration, but it suffers from very limited reconfigurability and modest frequency bandwidth (<6 GHz). Moreover, the frequency modulation of the laser diode itself requires an electronic AWG, so it is ultimately constrained by the specifications of available AWG instrumentation (e.g., limited bandwidth) while sharing the above-mentioned drawbacks of this solution.

In summary, all the previously reported photonic techniques either need the use of bulky and expensive ML lasers, electronic AWGs, require high levels of group-velocity dispersion through bulky fiber-based lines, and/or are limited in flexibility, or in terms of coherence performance. These drawbacks represent a fundamental hurdle towards the use of photonic RFCW for real-world applications.

Here, we propose a novel concept for the photonic generation of broadband linear RFCWs based on an extremely simple implementation, involving a single CW laser and standard low-frequency electronics, thus avoiding the need for ML or multiple lasers, long dispersive fiber lines, and fast electronics. The proposed scheme offers a time-bandwidth performance exceeding that of most photonic solutions and an extremely high degree of mutual coherence, while providing unprecedented flexibility in setting the RFCW specifications[32,33]. In detail, the fundamental parameters of the generated RF chirps (duration, chirp rate, etc.) are widely reconfigurable in real time by simply modifying the frequency of a MHz-range RF single tone. We report an experimental realization of the proposed concept in a fiber-optics platform, enabling generation of periodic RFCW trains with periods adjustable from 10 to over 100 ns, and with temporal durations that can be fully varied from zero to the period of the signal (from ~1 to 110 ns in the experiments reported here). In our demonstrated platform, the frequency bandwidth can potentially reach several hundreds of GHz, corresponding to a time-bandwidth product (TBP) above 2500. Additionally, we demonstrate electronic control of the chirp rate, which can be swept over its entire possible range, from positive to negative values, by simply tuning the input RF tone over a span of a few tens of kHz. Notice that obtaining equivalent chirp rates by linear dispersive propagation would require thousands of km of conventional single-mode fiber. Moreover, using an optical pulse shaper, the envelope of the generated RF chirps is easily controlled by linear optical spectral filtering, and the central frequency can be additionally tuned over tens of GHz. Finally, we demonstrate that the generated chirped waveforms are mutually coherent over durations exceeding 100 µs. All these properties make this concept very attractive for real-world applications in numerous fields, including detection and ranging (radar), microwave spectroscopy, and biomedical imaging, among many others.

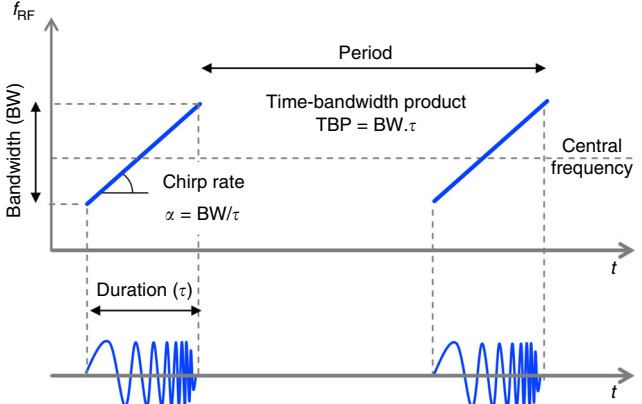

**Fig. 1** Schematic illustration of periodic linear RFCWs in the time—radio-frequency plane. Mathematically, an individual linear chirp can be described as the product between a slowly varying envelope function multiplied by a complex factor $e^{j\pi\alpha t^2}$, where $\alpha$ is called the chirp rate

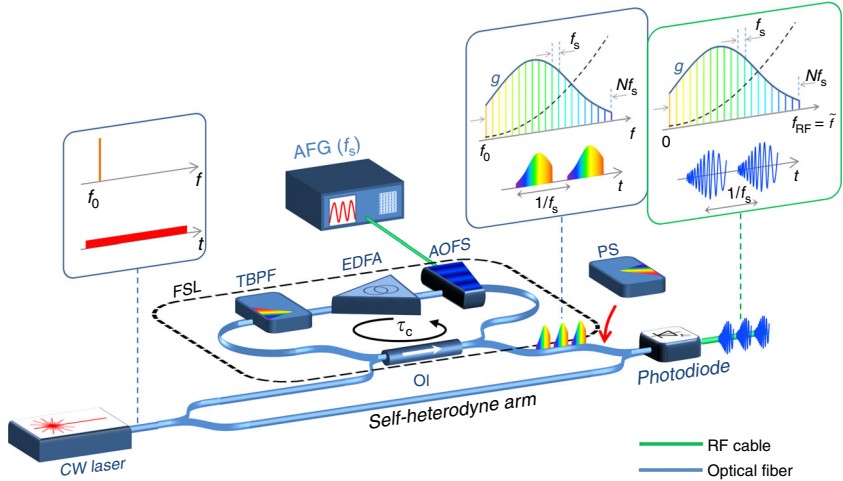

**Fig. 2** Schematic of the proposed photonic architecture for RFCW generation. A fiber loop (travel time: $\tau_c$) containing an acousto-optic frequency shifter (AOFS), an erbium-doped fiber amplifier (EDFA), a tunable bandpass filter (TBPF), and an optical isolator (OI), is seeded by a CW laser. The AOFS is driven by a single-tone RF signal at frequency $f_s$, produced from an arbitrary function generator (AFG). The FSL generates an optical frequency comb with a quadratic spectral phase (dashed line), corresponding in the time domain to a periodic train of time-stretched (dispersed) optical pulses. The optical waveform shape at the output of the loop can be controlled by a programmable optical spectrum pulse shaper (PS), and is then recombined with a fraction of the seed laser in a photodiode. The recombination with the seed laser produces the desired RF chirps by frequency down-conversion

## Results

**Electric field at the output of the frequency-shifting loop.** The reported platform for RF chirp generation is based on a single CW laser seeding a fiber recirculating frequency-shifting loop (FSL)[34,35]. The architecture of a CW-seeded FSL, depicted in Fig. 2, consists of an optical loop with an acousto-optic frequency shifter (AOFS) based on a Bragg cell, an optical bandpass filter, and an optical amplifier to compensate for the losses in the loop. The loop is seeded with a narrow-linewidth CW laser emitting at an optical frequency $f_0$, via an optical coupler. A second coupler allows us to extract a fraction of the optical field in the FSL.

The FSL is essentially described by two parameters: $\tau_c$, the travel time in the loop, and $f_s$, the frequency shift per round-trip (i.e., the frequency of the acoustic wave traveling in the AOFS). Notice that $f_s$ can be positive or negative, depending on the acousto-optic interaction in the Bragg cell. Owing to the successive round trips of the light in the frequency-shifting loop, the output optical power spectrum consists of a frequency comb, where the frequency of the $n$-th comb line is equal to $f_0 + n f_s$, with $n = 0, 1, 2\dots$ (note that the first line, at $n = 0$, corresponds to the CW seed laser). More precisely, defining $E_0(t) = E_0 e^{i2\pi f_0 t}$ as the electric field of the seed CW laser, the electric field at the output of the loop writes[35, 36]:

$$E(t) = E_0 \sum_n g(n|f_s|)e^{i2\pi(f_0+nf_s)t}e^{-i2\pi nf_0\tau_c}e^{-i\pi n(n+1)f_s\tau_c} \quad (1)$$

where $g(\tilde{f})$ is a real and positive function characterizing the envelope of the resulting spectrum, with $\tilde{f}$ being the positive baseband frequency, i.e., the optical frequency variable relatively to $f_0$ (in absolute value). $g^2(\tilde{f})$ is the power spectrum of the output field, and is set by the gain and losses in the FSL. We define $N$ as the largest integer for which $g(N|f_s|)>0$. In other words, $N|f_s|$ is the frequency bandwidth of the FSL comb.

Beside a linear term, the spectral phase in the expression of $E(t)$ (Eq. (1)) exhibits a quadratic dependence with the frequency variable ($nf_s$). This property is at the heart of a number of previously reported, unique properties of FSL systems, including the temporal Talbot effect[36, 37], the cancellation of intra-mode beatings in multimode lasers[38] and the recently demonstrated fractional Fourier transform in these systems[39].

We consider here the specific case when the product $f_s\tau_c$ is close to an integer value, and define $\Delta f_s = |f_s| - k/\tau_c$, where $k$ is a positive integer and $|\Delta f_s| << k/\tau_c$. For the sake of clarity, we show the details of the calculations in the case where $f_s > 0$ (i.e., $|f_s| = f_s$). Introducing the above expression for $f_s$ into Eq. (1) and neglecting the linear term ($\propto n$) in the spectral phase (as this term only translates into a change in the origin of times), the output electric field re-writes:

$$E(t) = E_0 e^{i2\pi f_0 t} \sum_n g(nf_s)e^{i2\pi nf_s t}e^{-i\pi n^2 \Delta f_s\tau_c}. \quad (2)$$

As detailed in the Methods, this electric field is identical to the field of a ML laser, with a spectral envelope $g$ and a free spectral range equal to $f_s$, that would have experienced a total group-delay dispersion (slope of the linear group delay as a function of radial frequency) equal to $D_2 = D$[19, 40], where:

$$D = \frac{\Delta f_s\tau_c}{2\pi f_s^2}. \quad (3)$$

Consequently, the output light field consists of a periodic repetition of temporally stretched optical waveforms, with a period equal to $1/f_s$. Notice that in the case where $\Delta f_s = 0$ (i.e., when the product $f_s\tau_c$ is an integer), all spectral components in Eq. (2) have the same phase, which ensures that optical pulses obtained at the output of the FSL are transform limited. This configuration has recently been used for real-time Fourier transformation of RF signals[41]. Here in the general case, it can be shown (see Methods) that provided that a temporal "far-field" condition corresponding to $|\Delta f_s|>2/(N^2\tau_c)$ is satisfied, the temporally stretched output waveforms have the following approximate expression:

$$E(t) \approx \frac{1}{\sqrt{|\Delta f_s|\tau_c}} E_0 e^{i2\pi f_0 t} \sum_n e^{\frac{i(t-n/f_s)^2}{2D}} g\left(\frac{t-n/f_s}{2\pi D}\right). \quad (4)$$

The output described by Eq. (4) corresponds to an infinite train of linearly chirped optical waveforms, with an optical chirp rate (slope of the instantaneous frequency change versus time) equal to $\alpha_{\text{opt}} = \frac{1}{2\pi D} = \frac{f_s^2}{\tau_c\Delta f_s}$. In the case where $f_s < 0$, calculations lead to

a similar expression of the individual output waveform, except for the fact that the complex chirp term in Eq. (4) should be replaced with $e^{-i\frac{(t-n/f_s)^2}{2D}}$. In this case, the optical chirp rate is then: $\alpha_{\text{opt}} = -\frac{f_s^2}{\tau_c \Delta f_s}$.

In both cases, the envelope of the output waveforms is the function $g$ mapped along the temporal domain, corresponding to the usual frequency-to-time mapping process that is encountered in dispersive propagation of short optical waveforms[8]. Notice that depending on the sign of $D$ (i.e., of $\Delta f_s$), the temporal envelope of the output waveform will map a direct or an inverted replica of the FSL spectrum, $g$. As such, provided that the output waveforms do not overlap, direct detection of the intensity at the output of the loop produces a photocurrent that maps repeatedly in time the power spectrum of the FSL:

$$I(t) = \frac{I_0}{|\Delta f_s|\tau_c} \sum_n \left[ g\left( \frac{t - n/f_s}{2\pi D} \right) \right]^2 \qquad (5)$$

where $I_0$ depends on both $E_0$ and the responsivity of the photodiode.

Let assume now that the optical signal at the output of the loop is recombined with a fraction of the seed CW laser (self-heterodyning), of amplitude $\varkappa E_0$. Then the photocurrent due to the contribution of an individual optical waveform is written:

$$I^{\text{SH}}(t) = |\varkappa E_0(t) + E(t)|^2. \qquad (6)$$

If the optical power of the seed laser in $N$th recombination arm is much higher than the optical power at the FSL output ($\varkappa \gg 1$), the resulting intensity is:

$$I^{\text{SH}}(t) \approx \varkappa^2 I_0 + \frac{2\varkappa I_0}{\sqrt{|\Delta f_s|\tau_c}} \sum_n g\left( \frac{t - n/f_s}{2\pi D} \right) \cos\left( \frac{(t - n/f_s)^2}{2D} \right). \qquad (7)$$

The result in Eq. (7) corresponds to an infinite train of periodic RFCWs, with a period equal to $1/|f_s|$, and an RF chirp rate equal to $\alpha_{\text{RF}} = \frac{1}{2\pi D} = \frac{f_s^2}{\tau_c \Delta f_s}$. Notice that the sign of the RF chirp rate only depends on the sign of $\Delta f_s$.

The envelope of the photocurrent associated with each individual chirped RF pulse corresponds to a temporal mapping of the spectral envelope of the FSL comb. When $\Delta f_s$ is positive ($D > 0$), the chirp rate is positive $\left( \alpha = \frac{f_s^2}{\tau_c \Delta f_s} \right)$, and the temporal envelope of the RFCW maps a direct copy of the spectral function $g$. In the other case ($\Delta f_s < 0$), the chirp rate is negative and the chirp envelope maps a reversed copy of $g$.

**Properties of the RF chirps**. To show the coherence properties of the waveforms, we express the photocurrent as defined in Eq. (6), by injecting the expression of the electric field given in Eq. (2) into Eq. (6). One obtains, regardless of the sign of $f_s$:

$$I^{\text{SH}}(t) \approx \varkappa^2 I_0 + 2\varkappa I_0 \sum_n g(n|f_s|) \cos\left(2\pi n|f_s|t - \pi n^2 \Delta f_s \tau_c\right). \qquad (8)$$

The signal detected by the photodiode consists of a sum of periodic harmonic functions with fundamental period $1/|f_s|$. This feature ensures the mutual coherence of successive RF waveforms. Notice that in practice, several factors can ultimately degrade the coherence of the output waveforms, specifically, the noise introduced by the EDFA, the stability of the loop and the recombination arm, the phase noise of the RF signal generator driving the AOFS, and the linewidth of the seed laser.

In the standard method for photonic-assisted RFCW generation, involving spectral pulse shaping and dispersive FTM, the optical

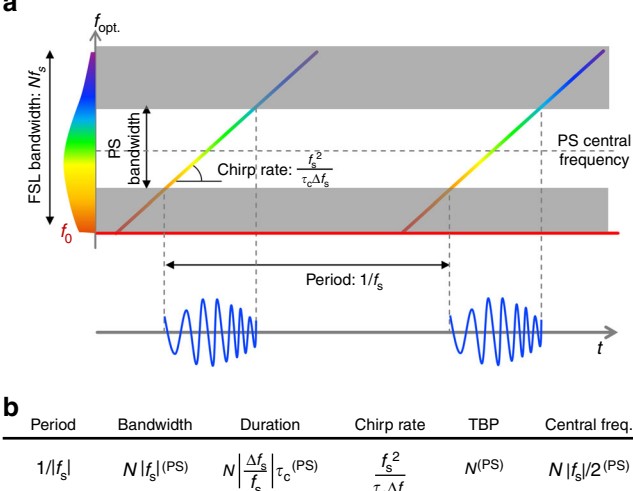

**b**

| Period | Bandwidth | Duration | Chirp rate | TBP | Central freq. |
|---|---|---|---|---|---|
| $1/|f_s|$ | $N|f_s|$(PS) | $N\left\|\frac{\Delta f_s}{f_s}\right\|\tau_c$(PS) | $\frac{f_s^2}{\tau_c \Delta f_s}$ | $N$(PS) | $N|f_s|/2$(PS) |

**Fig. 3** Control of the RFCWs parameters. **a** Plot in the time-optical frequency plane, of the optical chirps generated in the FSL (here: $f_s > 0$). Heterodyning the optical chirps with the seed laser (at $f_0$) produces the RF chirps. Both the central frequency and bandwidth of the chirps can be controlled by the pulse shaper (PS) at the output of the FSL. Here, the PS is used as a bandpass filter, and blocks the optical frequencies contained in the gray areas. **b** Summary of the RF chirps' parameters generated in the FSL as a function of the main design specifications of the FSL (see Fig. 2), i.e., round-trip propagation time in the FSL loop ($\tau_c$), in-cavity frequency shift ($f_s$), full optical bandwidth ($N|f_s|$), and differential frequency shift ($\Delta f_s = |f_s| - k/\tau_c$, where $k$ is a positive integer). Parameters marked with superscript (PS) can be subsequently adjusted by the optical pulse shaper (PS) at the output of the FSL

bandwidth is orders of magnitude larger than the RF chirp bandwidth[23]. In our case, both quantities are equal to $N|f_s|$. As such, the instantaneous frequency of the RF chirp varies from 0 to $N|f_s|$. As said above, the temporal envelope of the RF chirps is simply a time-mapped version of the square root of the power spectrum at the output of the loop. Owing to this intrinsic frequency-to-time mapping process, the temporal envelope of the generated RF chirped waveforms can be tailored by shaping the spectral envelope of the FSL optical output through a customized linear optical filtering device (e.g., a programmable spectral pulse shaper, PS). The latter can be placed at the output of the FSL, to provide additional control on the bandwidth, central frequency, and temporal and spectral envelopes of the generated RF chirped pulses (see Figs. 2 and 3a).

As shown in Eq. (7), the temporal envelope of the RF chirps maps the spectral envelope of the FSL comb. Then the temporal duration of the RF chirps $\tau$ is approximately equal to the spectral bandwidth of the FSL comb multiplied by the frequency-to-time mapping coefficient (group-velocity dispersion): $\tau = Nf_s \times 2\pi|D| = N\left|\frac{\Delta f_s}{f_s}\right|\tau_c$. Since the output signal is $(1/|f_s|)$-periodic, if $|\Delta f_s| > 1/(N\tau_c)$, two consecutive chirps will overlap, leading to the possibility of undesired interference among the RFCWs. As a result, $|\Delta f_s|$ should be kept smaller than $1/(N\tau_c)$. As a reference, the values of frequency shift introduced by a conventional AOFS range from a few tens to a few hundreds of MHz, which corresponds to temporal periods from a few ns to a few tens of ns.

The chirp rate in the RF domain is equal, in absolute value, to the chirp rate of the optical waveforms at the output of the FSL, i.e., inversely proportional to the equivalent dispersion introduced by the

FSL system (Eq. (3)), namely $\alpha_{RF} = \frac{f_s^2}{\tau_c \Delta f_s}$. The chirp rate is directly set by the value of $\Delta f_s$ and can be easily tuned by adjusting the frequency of the signal driving the AOFS (a MHz-range RF single tone). As said, the chirp rate can be either positive or negative, depending on the sign of $\Delta f_s$. As shown below, in our experimental platform, the absolute value of the chirp rate can be tuned over its entire range, from $N f_s^2$ to infinity ($N f_s^2$ corresponding to the maximum waveform duration of $1/|f_s|$), by simply sweeping the RF tone driving the AOFS over a span of a few tens of KHz.

Another critical advantage of the proposed configuration for RFCW generation is that it provides an equivalent group-velocity dispersion amount (Eq. (3)) that is significantly higher than that of standard optical dispersive lines. At this point, it is interesting to compare the equivalent dispersion introduced by the FSL with the typical dispersion values that can be achieved by conventional optical materials. Recall that a narrow-band pulse of light at a central wavelength of 1550 nm propagating through a 1-km long section of standard single-mode optical fiber (SMF-28) experiences a total group-velocity dispersion approximately equal to $D_{2,\,SMF} = -2 \times 10^{-23}\,s^2$ (anomalous dispersion regime). In the reported architecture, a characteristic value of the equivalent dispersion can be calculated, for example, in the limiting case when the duration of each output chirped waveform approaches the waveform train period, $1/|f_s|$, corresponding to $|\Delta f_s| = 1/(N\tau_c)$. Setting $\Delta f_s < 0$, the expression of the equivalent dispersion is then $D_{2,\,equiv} = -1/(2\pi N f_s^2)$. Considering typical values for a fiber-optics FSL such as the one demonstrated in our work, $N = 500$ and $f_s = 80$ MHz (i.e., a spectral bandwidth of 40 GHz), the equivalent group-delay dispersion is $D_{2,\,equiv} = -5 \times 10^{-20}\,s^2$. This corresponds to propagation through ~2500 km of SMF-28.

The time-bandwidth product (TBP) is often used as the standard figure of merit for chirped waveforms and it is defined as the product of the chirp duration by the chirp spectral bandwidth. In our proposed scheme, $\text{TBP} = N\left|\frac{\Delta f_s}{f_s}\right| \tau_c \times N|f_s|$. As discussed, the highest TBP is achieved when the individual waveform duration at the system output is set to coincide with the waveform period, namely when $|\Delta f_s| = 1/(N\tau_c)$. Recall that this ensures that overlapping is avoided among successive chirp waveforms. Therefore, the maximum TBP is simply equal to $N$, the number of frequency lines in the FSL comb. As shown in the following sections, using the proposed scheme, this value can be as high as a few thousands. Finally, all parameters of the RFCWs generated in the FSL are recalled in Fig. (3).

**Experimental results**. To demonstrate the capabilities offered by the proposed scheme for RFCW generation, we implemented two different fiber-optics FSL configurations (see Methods). The first one achieves RF chirps with a relatively short duration (<12 ns) and large bandwidth (>100 GHz). The second one is designed to generate long waveforms (>100 ns) with moderate bandwidth (<10 GHz). The components described in this section are represented in Fig. 2.

We characterize the spectral properties at the output of the FSL (configuration 1). First the bandwidth of the TBPF in the loop is set to 100 GHz, and the frequency shift per round-trip to $f_s = -77.185$ MHz. The loop round-trip time is $\tau_c = 77.735$ ns. The high-frequency edge of the filter (3 dB crossing point) is fixed to match the frequency of the seed laser. A 5 MHz resolution optical spectrum analyzer is used to monitor the optical spectrum of the frequency comb at the output of the FSL (Fig. 4a). Notice the excellent flatness of the comb (±3 dB over 1200 lines). The peak-to-noise ratio varies from 30 to 15 dB from the high to the low frequency range. Increasing the

bandwidth of the TBPF provides comb spectra as wide as 230 GHz (Fig. 4b), but at the expense of a degraded peak-to-noise ratio for lower frequency lines.

To characterize the spectral phase, and observe the quadratic variation predicted by Eq. (2), we evaluate experimentally the complex RF spectrum using the technique described below. First, we recombine the output of the FSL with the CW seed laser: the optical spectrum of the FSL comb is then down-converted to the RF domain. In order to limit the influence of intra-comb beatings, the optical power propagating through the reference arm is much higher than the power at the output of the FSL. A fast photodiode is used to measure the intensity of the corresponding signal, and a fast-Fourier transform is performed to obtain the power and the phase of the down-converted spectrum. Results are plotted in Fig. 4. These results clearly show the quadratic dependence of the FSL comb spectral phase, in agreement with the theoretical predictions in Eq. (2) and Eq. (8). The equivalent amount of group-delay dispersion in Fig. 4d ($f_s = -77.150$ MHz) and in Fig. 4h ($f_s = -77.220$ MHz) are equal respectively to $D_2 = -7.3 \times 10^{-20}\,s^2$ and $D_2 = +7.2 \times 10^{-20}\,s^2$. The first case is equivalent to the dispersion introduced by an ~3500-km long section of conventional single-mode fiber SMF-28. We reiterate that this technique allows creating the massive amounts of dispersion required for frequency-to-time mapping of nanosecond-long waveforms without need of any dispersive fiber line. Moreover, it is important to highlight that the tuning over this remarkable dispersion range (from positive to negative dispersion values) is achieved by sweeping the RF frequency tone driving the AOFS over a total span of 70 kHz only. Finally, when $f_s = -77.185$ MHz, the product $f_s \tau_c$ is an integer, which results in a flat (or linear) spectral phase (Fig. 4f). Under this condition, the FSL emits Fourier transform-limited pulses at a repetition rate equal to the comb FSR ($|f_s|$), similarly to a conventional ML laser[34, 35].

According to the previous theoretical calculations in Eq. (5), the intensity at the output of the FSL (i.e., before recombination with the seed laser) maps the optical power spectrum of the FSL comb. To verify this important feature, we adjust the spectral bandwidth to ~35 GHz by means of the TBPF and set $f_s = -77.353$ MHz, so that the product $\Delta f_s \tau_c = 1.9 \times 10^{-3}$ (here, $\tau_c = 77.591$ ns) (configuration 1). The signal detected by the photodiode at the output of the FSL maps the optical spectrum repeatedly in time (period: $1/|f_s| = 12.9$ ns) (Fig. 5a, b). After recombination with the seed laser, the envelope of the waveforms shows a similar shape, reproducing the square root of the optical power spectrum (Fig. 5c). Notice that the chirped waveforms (Fig. 5c) appear slightly shorter than the optical traces measured before recombination (Fig. 5a). This is due to filtering of high frequency components by the available detection scheme (28 GHz bandwidth, slightly smaller than 35 GHz, the bandwidth of the FSL comb) (Fig. 5d).

Finally, we show the RF power spectrum of a single chirped waveform (Fig. 5d). According to the theoretical expression of the chirped waveforms, the RF spectrum maps directly the optical spectrum of the FSL comb. As said, contrary to techniques combining pulse shaping and dispersive FTM, here the optical bandwidth is identical to the RF chirp bandwidth.

We now provide experimental evidence of reconfigurable generation of RFCWs with durations between 0 and 12 ns (configuration 1). The round-trip time is $\tau_c = 77.591$ ns. Two examples of RF chirps (obtained after heterodyning with the seed laser) are displayed in Fig. 6. As previously, the spectral bandwidth of the RF chirps is equal to 28 GHz, limited by the oscilloscope. To characterize the linearity of the generated RF chirps, we compute numerically the time-frequency energy distribution (Wigner–Ville) of the measured RFCW signals, i.e.,

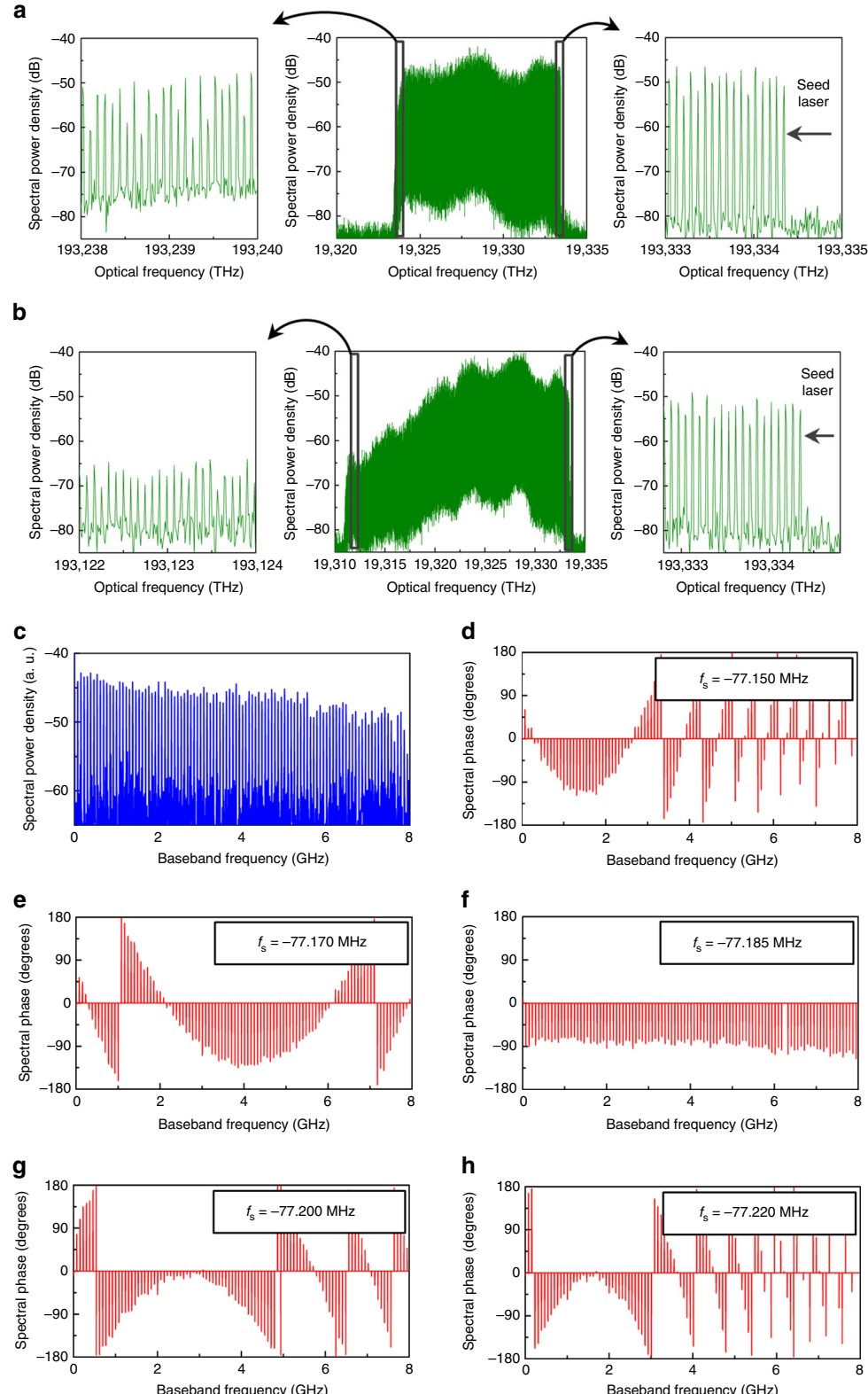

**Fig. 4** Spectral properties of FSLs. **a**, **b** Experimental optical spectra at the FSL output measured with a high-resolution optical spectrum analyzer (5 MHz resolution bandwidth). The spectral bandwidth of the measured waveforms is equal to ~100 GHz and ~230 GHz in **a** and **b**, respectively. **c** RF spectrum measured after recombination of the FSL comb with the CW seed laser (integration time: 1 μs). **d**–**h** Spectral phase (as a function of baseband frequency), measured for different values of the AOFS shifting frequency, $f_s$. f (flat spectral phase) corresponds to the situation when the product $f_s\tau_c$ is an exact integer (i.e., $\Delta f_s = 0$)

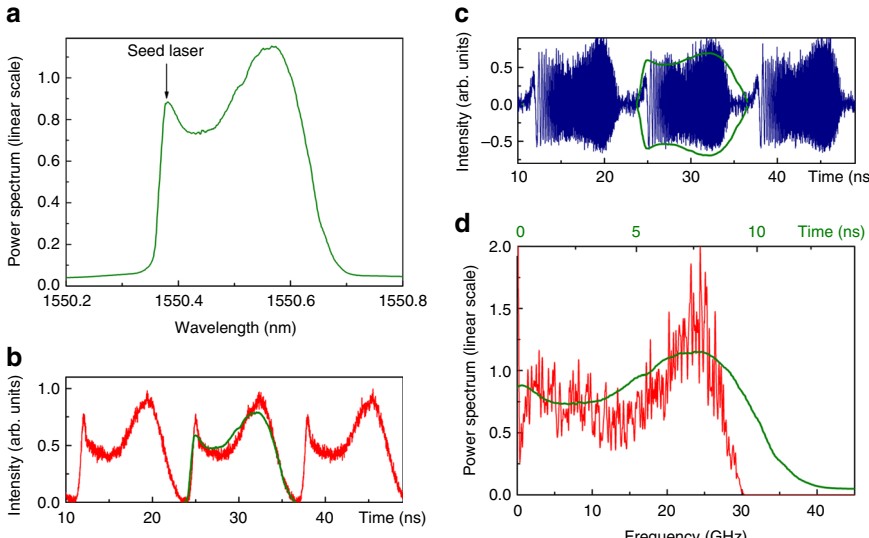

**Fig. 5** Evidence of frequency-to-time mapping at the output of the FSL. **a** Optical power spectrum generated in the FSL, in linear scale (the comb lines are not resolved). **b** In red: intensity measured at the output of the FSL, compared to the time-mapped optical power spectrum (green). **c** In blue: intensity measured after recombination with the seed laser, compared to an envelope profile mapping the square root of the optical power spectrum (green). **d** In red: spectrum of an individual chirped waveform extracted from **c**, in linear scale, compared to the time-mapped optical power spectrum (green). The spectral traces represented in **b**–**d** (green curves) are mapped into the time domain following Eq. (5), where the effective dispersion $D$ is calculated from Eq. (3) using the defined operation parameters, namely $=5.06 \times 10^{-20}\,\mathrm{s^2}$. All traces are single-shot (no averaged) measurements

the modulus in the time-frequency $(t, f)$ plane, of the function defined as[42]:

$$W(t,f) = \int I\left(t+\frac{\tau}{2}\right) I\left(t-\frac{\tau}{2}\right) e^{-i2\pi f\tau} \mathrm{d}\tau. \qquad (9)$$

As compared to other standard time-frequency representations, such as the windowed Fourier transform, the Wigner–Ville distribution enables a much better resolution in the time-frequency plane. It is thus especially suited for single linearly chirped signals[43]. Here, the Wigner–Ville density plots demonstrate the excellent quality and linearity of the experimentally generated RF chirps (Fig. 6).

To demonstrate the potential of our newly proposed technique for generation of RF chirps with arbitrary central frequency, chirp rate and sign, we perform measurements with different values of

the AOFS shifting frequency (configuration 1). In all the experiments reported herein, the central wavelength is set by the pulse shaper (PS, Fig. 2) to 1550.1 nm (193.5 THz), and the bandwidth is set equal to 0.1 nm (12.5 GHz). The obtained time traces, measured after recombination with the seed laser, and the corresponding Wigner–Ville density plots are shown in Fig. 7. As can be seen, tuning of the RF chirps from $+2.5 \times 10^{+18}\,\mathrm{s^{-2}}$ to $-2.5 \times 10^{+18}\,\mathrm{s^{-2}}$ is obtained by simply sweeping the AOFS frequency over a full span of ~64 kHz. Notice that the "ghost" components appearing in some time-frequency plots (e.g., in Fig. 7k) are only due to numerical artefacts occurring in the computation of the Wigner–Ville distribution of signals with non-zero average value, and do not represent actual frequency components.

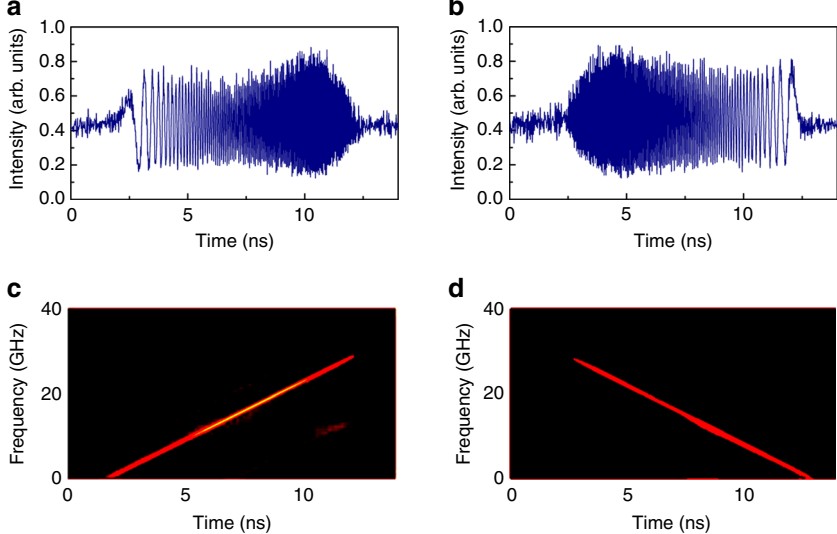

**Fig. 6** Examples of broadband arbitrary RFCWs. RF chirps generated with $f_s = -77.353$ MHz (**a**, **c**) and $f_s = -77.303$ MHz (**b**, **d**). **a**, **b** Temporal traces recorded by the fast photodiode; **c**, **d** numerically calculated Wigner–Ville distributions of the measured RF waveforms

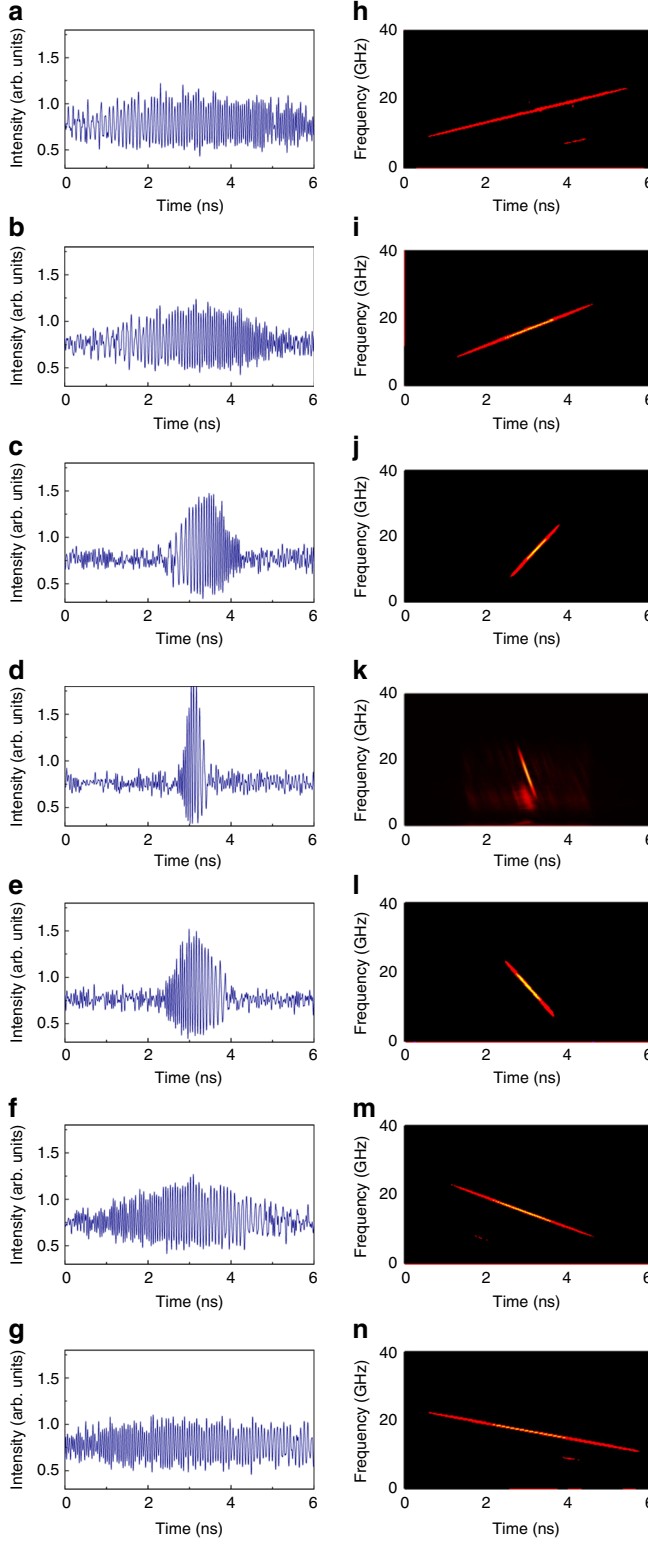

**Fig. 7** Arbitrary tuning of the RF chirp rate as a function of $f_s$. The central frequency in all the generated chirp pulses is set to ~15 GHz, and the spectral bandwidth is set to ~12 GHz. The corresponding values of $f_s$ (in MHz) for **a**–**f** and **g** are, respectively, equal to −77.281, −77.271, −77.260, −77.251, −77.248, −77.235, and −77.217. **h**–**n** Corresponding Wigner–Ville distributions

In this section, we first demonstrate arbitrary envelope shaping of the RF chirps (configuration 1). We set $f_s$ =−77.303 MHz ($\tau_c$ =77.591 ns). Through the use of a commercial programmable linear optical spectral shaper (PS, see Fig. 2), we apply successively a band-stop filter (central wavelength: 1550.1 nm, spectral bandwidth: 0.1 nm), and a bandpass filter (central wavelength: 1550.1 nm, 0.1 nm bandwidth).

The RF waveforms detected after recombination with the seed laser, shown in Fig. 8a, b, follow a temporal shape that resembles the spectral amplitude response that is programmed in the pulse shaper. In another group of experiments, we demonstrate arbitrary tuning of the central frequency and bandwidth of the generated RF chirps, by controlling the central wavelength of a bandpass optical filter (0.1 nm bandwidth) implemented through the PS. In Fig. 8c–j, we plot the RF waveforms obtained when the central wavelength of the PS is set at 1550.0 nm (Fig. 8c, e), 1550.1 nm (Fig. 8d, f), 1550.2 nm (Fig. 8g, i), and 1550.3 nm (Fig. 8h, j). The central frequencies of the RFCW are, respectively, equal to ~3 GHz, ~12 GHz, ~24 GHz, and ~27 GHz (limited by the detection bandwidth).

Then, we demonstrate the mutual coherence of the emitted RF chirped waveforms. To do so, the FSL is set in configuration 2, where two AOFS producing frequency shifts with opposite signs are inserted in the FSL. The FSL round-trip time is now set to $\tau_c$ = 89.71 ns. The net frequency shift per round-trip is positive and set to $f_s$ = 11.147 MHz. The spectral bandwidth is 3.5 GHz (limited here by the oscilloscope bandwidth) and the duration of individual chirped waveforms is ~55 ns. A ~100 μs-duration time trace is recorded (corresponding to ~1110 consecutive waveforms) (Fig. 9a). An individual trace is numerically extracted from the sequence, and used as a reference, to calculate the cross-correlation of all individual waveforms[44]. The cross-correlation trace displayed on Fig. 9b, shows no attenuation, nor broadening of the correlation peaks, as compared to the auto-correlation peak (at null delay), demonstrating the extremely high degree of coherence of the generated RF chirps (Fig. 9c, d). The width (FWHM) of the central peak of the auto-correlation function is $\tau_0$ ~ 290 ps. The compression ratio, defined as the ratio between the chirp duration and $\tau_0$, is found to be equal to ~190. As expected, this value is very close to the TBP allowed by our detection system, i.e., ~192.

Finally, we report the generation of long (>100 ns duration) broadband RF chirps. The FSL is set in configuration 2, the net frequency shift per round-trip is positive and set close to $f_s = 1/\tau_c$ = 8.569 MHz, and the spectral bandwidth of the TBPF is ~10 GHz. The detection bandwidth is 25 GHz. The results displayed in Fig. 10 demonstrate the possibility of generating arbitrary RF chirps, with duration exceeding 110 ns, and a bandwidth of 10 GHz, corresponding to a time-bandwidth product larger than 1000.

## Discussion

We have proposed and experimentally validated a novel and extremely simple concept for photonic generation of arbitrary broadband RF waveforms with linear frequency modulation (or chirps). Contrary to the vast majority of the approaches reported so far, which typically rely on ML or multiple lasers, bulky dispersive lines and/or fast electronics, our scheme makes use of a simple fiber-optics loop seeded by a CW laser and controlled by standard low-frequency electronics. This represents a considerable improvement in terms of reduced system complexity, size,

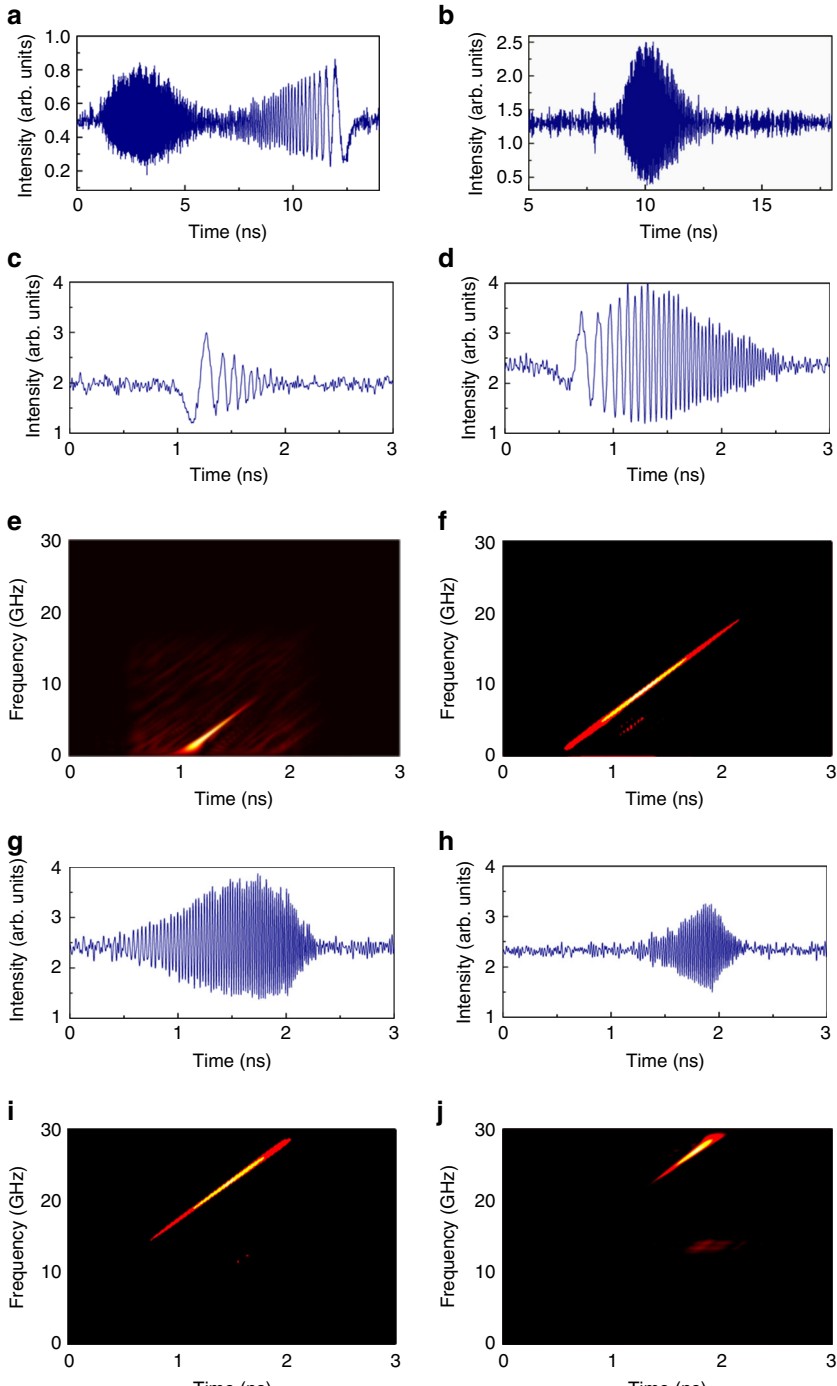

**Fig. 8** Temporal and spectral shaping of the RF chirps by optical spectral shaping of the FSL comb. Two different spectral amplitude masks are used, i.e., **a** band-stop filter in **a** and a bandpass filter in **b**. **c–j** Control of the central frequency of the RF chirps by tuning the central wavelength of the PS used as a tunable bandpass filter (0.1 nm bandwidth) at the output of the FSL. The RF waveforms and the corresponding Wigner–Ville distributions are shown for different values of the central wavelength of the bandpass filter (**c**, **e** 1550 nm; **d**, **f** 1550.1 nm; **g**, **i** 1550.2 nm, and **h**, **j** 1550.3 nm). The detection bandwidth is 28 GHz

weight and the associated potential cost, as well as in terms of increased stability and flexibility, all of them critical features for real-world applications.

The proposed architecture directly generates a comb of optical frequencies with an arbitrary quadratic spectral phase, therefore equivalent to a ML laser affected by a reconfigurable, user-defined (and possibly ultrahigh) amount of group-velocity dispersion. The time-frequency performance of this method is at least comparable to state-of-the-art photonic-based techniques, but the

scheme provides an unprecedented flexibility to tailor the key specifications of the chirped waveforms. The RF chirps can be arbitrarily and dynamically tailored in central frequency, bandwidth, time duration, and sign of the chirp by simply tuning a MHz-range RF single-frequency tone; additional control on the temporal envelope shape and other waveform features can be achieved through widely available optical spectral shaping. The results reported here are exclusively limited by the available photo-detection bandwidth. By simply employing a faster

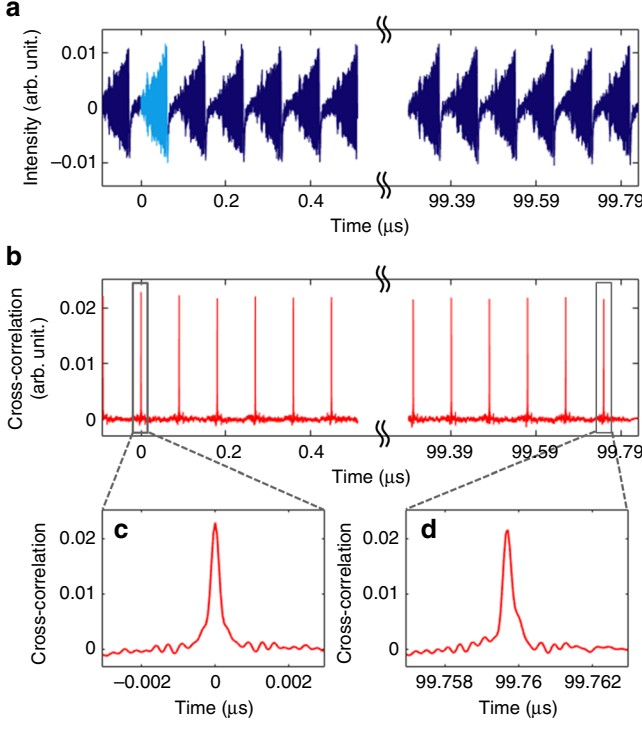

**Fig. 9** Mutual coherence of RFCWs. **a** Sequence of ~1110 consecutive measured RFCWs used for numerical calculation of the temporal cross-correlation trace. The reference waveform is plotted in light blue. **b** Cross-correlation trace. The auto-correlation trace (correlation of the reference with itself) corresponds to null delay time. **c**, **d** Comparison of the auto-correlation peak (**c**), with the cross-correlation signal obtained for a delay of ~100 μs, i.e., between the reference waveform (#1), and the waveform #1112 (**d**)

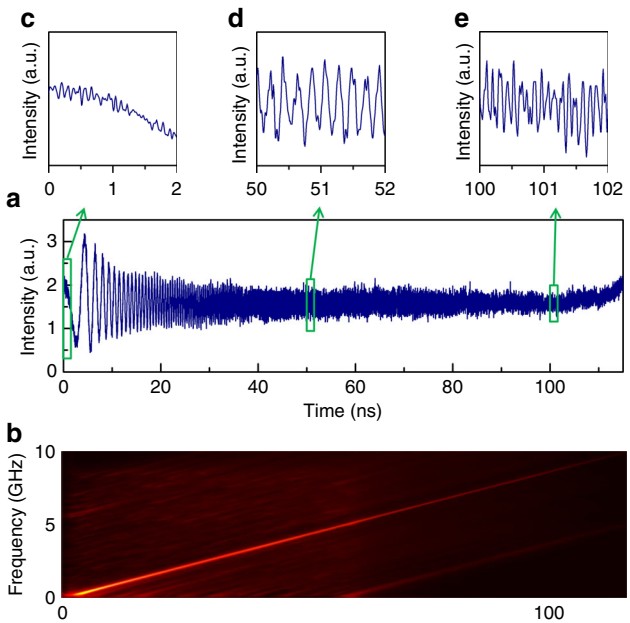

**Fig. 10** Demonstration of a broadband record-long RFCW. In this experiment, $f_s = 8.5491$ MHz. The complete RFCW is plotted in **a**. **b** corresponding Wigner–Ville distribution. The instantaneous frequencies at 0, 50, and 100 ns (temporal traces shown in the insets labeled with **c**, **d**, and **e**) are, respectively, equal to DC, 4.5 GHz, and 9 GHz

photodetector, e.g., a UTC-PD, the fiber-optics platform demonstrated in our work would enable generation of RF chirped waveforms with a bandwidth above the 200-GHz mark. Moreover, the RF waveforms are generated in a periodic fashion and are mutually highly coherent, a characteristic of critical importance for many practical applications, such as radar compression by matched filtering[1, 2]. For example, the coherence time of the RFCW reported here exceeds 100 μs, which would enable radar coherent detection at a range exceeding tens of kilometers. More generally, the architecture of the FSL could be optimized according to the specifications of the targeted application. As said earlier, the duration of the RFCW is set by $f_s$ and $\tau_c$ (the maximum duration is limited by the period of the waveforms, i.e., the smallest value between $\tau_c$ and $1/|f_s|$). In our case, we have demonstrated waveforms durations in the 10 to 100 ns range. This value could be further enlarged by increasing the value of $\tau_c$ (i.e., using a longer FSL loop). For example, implementing a FSL with a round-trip propagation time of 1 μs, and a frequency shift of 1 MHz (a value that could easily be obtained by using two AOFS with opposite signs, such as in configuration 2), would enable the generation of RFCWs with a duration in the μs range. On the other hand, generating shorter waveforms would be possible by increasing the value of $f_s$. Since the frequency shifts provided by the AOFS are limited to a few hundreds of MHz, the use of single-sideband electro-optic modulators would be an alternative solution to provide frequency shifts per round-trip well above the GHz range[45]. In this case, the bandwidth of the RFCW could be considerably increased, up to the THz range. Based on the demonstrated performance and future potential upgrades, the demonstrated concept represents a significant advancement towards the development of practical broadband RF chirped waveform generation systems with specifications commensurate with those needed for real-world applications in numerous fields, including radar-based detection and ranging, microwave spectroscopy and biomedical imaging.

## Methods

**Phase-locked pulse train in a linear dispersive medium.** Consider a phase-locked pulse train, defined by the complex amplitude:

$$E_{in}(t) = E_0 e^{i2\pi f_0 t} \sum_n g(nf_s) e^{i2\pi n f_s t} = E_0 e^{i2\pi f_0 t} \mathcal{E}_{in}(t) \quad (10)$$

where $\mathcal{E}_{in}(t)$ is the envelope of the pulse train. The corresponding spectrum is a Dirac comb:

$$\tilde{E}_{in}(f) = \int E_{in}(t) e^{-i2\pi ft} dt = E_0 \sum_n g(nf_s) \delta(f - f_0 - nf_s). \quad (11)$$

Suppose that the pulse train propagates through a linear dispersive medium, exhibiting group-velocity dispersion. We define the group delay $\tau_d$ as the product of $\beta'$ (i.e., the first derivative of the propagation constant $\beta$ with respect to the angular frequency at $\omega_0 = 2\pi f_0$), by the propagation distance $L$[40]. Similarly, we define the group-velocity dispersion (GDD) $D_2$, as the product of $\beta''$ (second derivative of $\beta$ with respect to the angular frequency at $\omega_0$), by the propagation distance $L$. After a propagation distance $L$, the pulse has acquired a quadratic spectral phase, and the spectrum re-writes[40]:

$$\tilde{E}_{out}(f) = E_0 \sum_n g(nf_s) \delta(f - f_0 - nf_s) e^{-i2\pi \tau_d (f-f_0)} e^{-i2\pi^2 D_2 (f-f_0)^2}. \quad (12)$$

The complex amplitude at the output of the dispersive medium is given by:

$$E_{out}(t) = \int \tilde{E}_{out}(f) e^{i2\pi ft} df \propto E_0 e^{i2\pi f_0 t} \sum_n g(nf_s) e^{i2\pi n f_s (t-\tau_d)} e^{-i2\pi^2 n^2 D_2 f_s^2}. \quad (13)$$

Formally, this expression is identical to that of Eq. (2), provided that:

$$D_2 = \frac{\Delta f_s \tau_c}{2\pi f_s^2}. \quad (14)$$

**Electric field generated at the output of the FSL**. Owing to the equivalence between the electric field at the output of the FSL, and a transform-limited ML pulse train (envelope: $\mathcal{E}_{in}(t)$) after propagation through a linear dispersive medium exhibiting group-velocity dispersion, we can formally write the envelope of the electric field at the output of the FSL $\mathcal{E}(t)$, as the convolution integral of $\mathcal{E}_{in}(t)$, with the impulse response of the dispersion operator, $h_T(t) = \frac{1}{\sqrt{i2\pi D_2}} e^{i\frac{t^2}{2D_2}}$[40].

Then:

$$\mathcal{E}(t) = [h_T * \mathcal{E}_{in}](t) = \frac{1}{\sqrt{i2\pi D_2}} \int \mathcal{E}_{in}(t') e^{i\frac{(t-t')^2}{2D_2}} dt'. \quad (15)$$

The Poisson summation formula writes:

$\mathcal{E}_{in}(t') = \sum_n g(nf_s) e^{i2\pi nf_s t'} = \frac{1}{f_s}\sum_n G\left(t' - \frac{n}{f_s}\right)$, where $G(t) = \int g(\tilde{f}) e^{i2\pi \tilde{f} t} d\tilde{f}$ is the inverse FT of $g(\tilde{f})$. Simple mathematics lead to:

$$\mathcal{E}(t) = \frac{1}{f_s \sqrt{i2\pi D_2}} \sum_n e^{i\frac{(t-n/f_s)^2}{2D_2}} \int G(t'') e^{i\frac{t''^2}{2D_2}} e^{-i\frac{(t-n/f_s)t''}{D_2}} dt''. \quad (16)$$

The duration of $G(t'')$ is determined by the spectrum of the FSL (i.e., by the bandwidth of $g(\tilde{f})$), and is of the order of $1/(Nf_s)$. The far-field approximation corresponds to the case when $|D_2|$ is large enough so that the phase variation $\left|\frac{t''^2}{2D_2}\right|$ is smaller than $\pi/8$ over the duration of the transform-limited pulse, $1/(Nf_s)$[20, 23]. This condition writes: $|\Delta f_s| > 2/(N^2 \tau_c)$. If satisfied, the variation of the term $e^{i\frac{t''^2}{2D_2}}$ in the integral can be neglected, which leads to the following expression:

$$\mathcal{E}(t) \approx \frac{1}{f_s \sqrt{i2\pi D_2}} \sum_n e^{i\frac{(t-n/f_s)^2}{2D_2}} g\left(\frac{t - n/f_s}{2\pi D_2}\right). \quad (17)$$

Finally, neglecting the phase term $1/\sqrt{i}$ (resp. $1/\sqrt{-i}$) when $\Delta f_s > 0$ (resp. $\Delta f_s < 0$), the electric field at the output of the FSL writes:

$$E(t) = E_0 e^{i2\pi f_0 t} \mathcal{E}(t) \approx \frac{E_0}{\sqrt{|\Delta f_s|\tau_c}} e^{i2\pi f_0 t} \sum_n e^{i\frac{(t-n/f_s)^2}{2D}} g\left(\frac{t - n/f_s}{2\pi D}\right) \quad (18)$$

where we have made use of the equality $D_2 = D$.

**First experimental set-up**. In its first configuration, the set-up consists of a single-mode fiber loop containing an erbium-doped fiber amplifier (EDFA), a tunable bandpass filter (TBPF), and a fiber acousto-optic frequency shifter (AOFS) driven around 80 MHz (±5 MHz). The AOFS induces a negative frequency shift ($f_s < 0$). The AOFS insertion losses are close to 2 dB. The loop also contains an optical isolator to prevent back-reflection. The role of the TBPF is twofold: first to adjust the spectral bandwidth of the comb, and second, to limit the influence of the amplified spontaneous emission from the EDFA. The travel time of light in the loop is $\tau_c \sim 77.7$ ns. A narrow-linewidth (<0.1 kHz) CW laser, delivering 2 mW of optical power at 1550.0 nm is split by means of a 3 dB coupler (Fig. 2). One of the outputs is attenuated and used to seed the FSL by means of a 2% Y-coupler. The polarization of the seed laser is controlled by a fiber-pigtailed polarization controller. A 1% Y-coupler extracts a fraction of the light circulating in the loop (the optical power at the output of the coupler is between 10 and 100 µW). After passing through an optical pulse shaper (PS) with a 0.1 nm spectral resolution, the output light is recombined with the seed laser after the second output port of the 3 dB coupler. In this configuration, the repetition period of the generated waveforms is close to 12 ns.

Finally, in our experiments, the intensity after recombination is detected by a fast photodiode (20 ps risetime), and recorded by a 28 GHz- or a 25 GHz-bandwidth real-time digital oscilloscope. It is important to note that all measured waveforms reported here are single-shot time traces (no averaging is performed).

**Second experimental set-up**. A second implementation of the seeded FSL has been designed for the generation of long RF waveforms (100 ns). Depending on the fiber components inserted in the loop, the value of $\tau_c$ is in the range of 90-120 ns. To increase the temporal period of the output RF waveforms (i.e. to reduce the value of $f_s$), two AOFS are inserted in the loop and provide frequency shifts around 80 MHz, with opposite signs. The resulting frequency shift $f_s$ can easily be set between −12 and +12 MHz. The total insertion losses of the cascaded AOFS devices are ~4 dB. This second configuration relies on the same architecture as the first one, but is implemented with polarization-maintaining (PM) fiber components, avoiding the need for polarization control. The spectral bandwidth of the TBPF is set to ~10 GHz. The photodiode output current is detected by a 3.5 GHz, or by a 25 GHz analog bandwidth oscilloscope.

**Data availability**. The data that support the findings of this study are available on request from the corresponding author H.G.d.C.

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

## Acknowledgements

This research was supported by the Natural Science and Engineering Research Council (NSERC) of Canada and the Fonds de recherche du Québec—Nature et technologies (FRQNT), by the Région Rhône-Alpes (Explora' Pro program, Grant Number 14.004453), by the Agence Nationale de la Recherche (Grant Number ANR-14-CE32-0022), and by the Institut National de la Recherche Scientifique (INRS). We acknowledge Tektronix France for the loan of a 25 GHz-bandwidth real-time digital oscilloscope.

## Author contributions

H.G.d.C. and J.A. initiated the project, H.G.d.C., L.R.C., C.S., and M.B. carried out the experiments, C.S. and H.G.d.C. carried out the theoretical analysis. All authors contributed to the manuscript writing. H.G.d.C. and J.A. provided management oversight for the project.

## Additional information

**Competing interests:** The authors declare no competing interests.

