## [Peer Review File · Nature Communications]

Reviewers' comments:

Reviewer #1 (Remarks to the Author):

The paper presents a method for photonic generation of large bandwidth linearly chirped waveforms with good reconfigurability on sign and value of the chirp rate, central frequency and bandwidth, chirp duration and repetition rate. The work is interesting, but I do not think it meets the criteria for publication in nature communications. The reasons are listed below.

1. The authors claimed that the proposed scheme offers a time-bandwidth performance exceeding that of state-of-the-art photonic solutions. I do not think so. In the literature, there were many photonic methods reported to generate better performance RF chirped waveforms using only low-speed electronics or even no RF frequency input. For instance, [r1, r2] reported photonic generation of linearly chirped waveforms with time-bandwidth products (TBPs) of more than 100000 and good reconfigurability.
2. In practice, TBP around 2500 is too small for almost all the applications using RF chirped waveforms. From the description in the paper, the TBP limitation is hard to be broken. Therefore, the practical application of the proposed scheme, even after optimization of the parameters, is difficult.
3. Pulse integration or accumulation is a fundamental technique to increase the sensitivity of systems applying RF chirped waveforms, which requires the coherence of the pulse train. The work applied autocorrelation or cross-correlation of very few pulses to evaluate the coherence between the pulses, which does not make sense. One method to do so is to do the pulse accumulation for tens of pulses and see if there is accumulation gain. In addition, from the measured cross-correlation coefficient of 96%, 93%, 87%, the coherence of the generated chirped pulse train is poor.
4. The authors should check equation (1). To make equation correct, I think the AOFS should be placed before the EDFA, otherwise, another parameter should be defined to represent the time delay from the AOFS to the OI and should be presented in (1).
5. The authors should check the experiment parameters. For instance, $\Delta f \times \tau_c$ calculated from the experiment parameters does not equal to the results you provided in the paper.

[r1] J. M. Wun, C. C. Wei, J. Chen, C. S. Goh, S. Y. Set, and J. W. Shi, "Photonic chirped radio-frequency generator with ultra-fast sweeping rate and ultra-wide sweeping range," *Opt. Express* 21(9), 11475–11481 (2013).

[r2] P. Zhou, F. Z. Zhang, Q. S. Guo, and S. L. Pan, "Linearly chirped microwave waveform generation with large time-bandwidth product by optically injected semiconductor laser," *Opt. Express*, 24(15), 18460-18487 (2016).

Reviewer #2 (Remarks to the Author):

The manuscript describes a new microwave photonic solution for programmable RF chirped waveforms generation. This topic has been extensively studied in recent microwave photonic research. Most existing photonic approaches rely on the use of either broadband pulsed laser or high-speed electronics. The approach demonstrated in this work is conceptually different from the existing methods in such a way that only simple and low-cost cw laser and low-speed electronics are required. This is made possible by replacing large dispersion stretching with an optical frequency comb with quadratic phase profile generated from a frequency shifting loop. This concept has been successfully employed by the authors' group in various applications. Here it is reported the use of this concept for reconfigurable chirped RF waveform generation. The proposed method is well presented in general and verified by comprehensive experimental demonstrations. Results are technically sound.

I am convinced that the proposed approach offers essential benefits compared to some existing photonic methods reported in [12-17] for chirped RF waveform generation. Generation of various chirped RF waveforms is import, which has been demonstrated in this work. The capability of arbitrary waveform generation is also highly demanded and has been widely achieved by other approaches. Short of this capability may limit its significance. Comments on this issue will help the readers better understand this method.

Some technical comments:

Abstract:

- "state-of-the-art photonic techniques ... limited tuning performance". This is not quite accurate. Many existing methods offer reconfigurable/programmable capability for arbitrary waveform generation. Some relevant references should be cited.

- "a wide range of real-world applications." This statement is too generic. Some solid examples should be identified.

Introduction:

- "corresponding to the equivalent of dispersive propagation through thousands of km of conventional single-mode fiber". Dispersive propagation is not clear. Does it particularly refer to the proposed method or existing ones, such as frequency-to-time mapping?

Theoretical Description:

- The overall shape of optical comb spectrum $g(\omega)$ is an important parameter. How it can be determined and designed?
- More information is needed to support the statement of Fourier transform-limited output. Why an integer product of f_{osc} guarantees this?

Experimental Results:

- Are the waveforms shown in Fig. 5b,c single-shot measurement or averaged results?
- In Fig. 5d, green curve shows the time-mapped optical power spectrum. Time scale should be given.
- Caption of Fig. 5, the first b should be a.
- In section D, an optical spectral shaper was used to tailor the temporal envelope. What is the spectral resolution of this shaper?
- the acousto-optic frequency shifter is a key component to generate optical frequency combs. Two are used in configuration #2. What is the diffraction efficiency or the overall insertion loss of this device in the loop?

Reviewer #3 (Remarks to the Author):

The authors report a novel technique for generating broadband RF chirped pulses.

This technique has a salient number of features:

- a) It employs a simple CW laser rather than a mode-locked source
- b) It can be fully reconfigured in spectrum and in central frequency and/or chirp parameter and sign by means of a low frequency modulation source and a tunable optical filter
- c) It does not require a broadband source

- d) It requires a photodetector with a bandwidth compatible with that of the RF chirped pulse
- e) Have a time bandwidth product several orders of magnitude better than other current available approaches

The system is based on exploiting the inclusion of an acousto-optic tunable Frequency shifter and a tunable optical filter inside a loop. The input CW signal to the loop is actually transformed into the equivalent of a comb source by successive recirculations and the optical filter tailors the output pulse shape. Depending on the detuning between the frequency modulation the acousto-optic frequency shifter and the inverse of the cavity delay either a positive/negative or transform-limited periodic pulse train is generated. The authors then demonstrate experimentally the concept and provide results that show tunability of the chirp parameter and sign, central frequency etc. Two examples are illustrated featuring different tradeoffs between delay and bandwidth.

The concept to me is original and the idea is certainly excellent. My concern is that this paper, as it is written is quite difficult to follow, not only to the expert in the field, but more precisely to the general reader of Nature Communications

Some derivations in the paper require a more detailed development. For instance those leading to Eq(3), that leading to the expression of the temporal duration in section B.3, and Eq(13) in Methods. There is also some terminology that should be explained, which might not be straightforward to the non-specialized reader. For example, what is the meaning of the "individual output waveforms"? The authors could try, for example to include an expression of the total field in terms of the sum of individual contributions, to make this picture more clear. Regarding terminology, the authors use either ring or loop adjectives. To me a fiber loop is a Sagnac and not a ring cavity..

All in all, the main problem that I see is that the model includes a considerable number of parameters and it is not easy to see how manipulating one or various of them result in the changing of one or different features of the Chirped RF pulses. I suggest that the authors draft a sort of "design table" where they can put in the first column the relevant design parameters (τ_c , F_s , Δf_s , etc...) and in the first row the relevant features to be reconfigured (chirp rate, central frequency, pulse bandwidth, pulse shape etc...). This would make things much more clear. On another table that could include some information regarding realistic values of these parameters to achieve certain ranges of given parameters and for which application they might suit. I give an example: a certain range of bandwidth and delays might be useful for instance for SAR Radar, other could be useful for some bio application and so on.

These two last suggestions could also illustrate and support the text of a much missed discussion section.

Reviewer #1 (Remarks to the Author):

1. The authors claimed that the proposed scheme offers a time-bandwidth performance exceeding that of state-of-the-art photonic solutions. I do not think so. In the literature, there were many photonic methods reported to generate better performance RF chirped waveforms using only low-speed electronics or even no RF frequency input. For instance, [r1, r2] reported photonic generation of linearly chirped waveforms with time-bandwidth products (TBPs) of more than 100000 and good reconfigurability.

We thank the referee for pointing these two references. They have been added to the bibliography. A point that was not clear enough in the submitted manuscript is the fact that our scheme generates trains of RF chirped waveforms, with an extremely high degree of mutual coherence. To emphasize this point, we ran additional experiments, to show that the train of RF chirps shows no noticeable loss in coherence over a duration of 100 μ s, corresponding to more than 1100 consecutive waveforms. These results have been summarized in Fig. 9. This coherence property is crucial for many of the practical applications of RF chirped waveforms, such as for coherent radar applications (FMCW, pulse compression), and for pulse accumulation. Moreover, it constitutes a fundamental advantage of our concept, particularly over sources of RFCW based on mixing of two different lasers, such as the ones mentioned by the referee. In this case, without relative stabilization of the two lasers, the RFCW carry the intrinsic phase and frequency noise of the two lasers, resulting in a poor degree of temporal coherence of the train of chirps. Hence, we agree that two-lasers techniques can exhibit a larger TBWP, but would be of limited practical use. We have modified the introduction section, so as to underline the importance of the mutual coherence of the chirp train. To take the referee's remark into account, we changed the statement to: "a time-bandwidth performance exceeding that of most photonic solutions".

2. In practice, TBP around 2500 is too small for almost all the applications using RF chirped waveforms. From the description in the paper, the TBP limitation is hard to be broken. Therefore,

the practical application of the proposed scheme, even after optimization of the parameters, is difficult.

We thank the referee for bringing up this point, but disagree with his/her statement that all applications using RF chirped waveforms will require a TBP beyond that demonstrated in our work (~2500). For instance, in radar applications, the compression factor is equal to the TBP, and as such, a compression factor of 2500 enables a considerable gain in timing/distance resolution. In fact, even TBP products of a few tens are commercially used in air surveillance radars (e.g. AN/FPS-117, see: <http://www.radartutorial.eu/09.receivers/rx53.en.html>). On the other hand, we are currently investigating the intrinsic limitations of the combs produced in an FSL. We have now been able to obtain more than 5000 lines, which tends to show that the TBWP limit of 2500 we give in the manuscript could be significantly extended.

3. Pulse integration or accumulation is a fundamental technique to increase the sensitivity of systems applying RF chirped waveforms, which requires the coherence of the pulse train. The work applied autocorrelation or cross-correlation of very few pulses to evaluate the coherence between the pulses, which does not make sense. One method to do so is to do the pulse accumulation for tens of pulses and see if there is accumulation gain. In addition, from the measured cross-correlation coefficient of 96%, 93%, 87%, the coherence of the generated chirped pulse train is poor.

This is a very interesting point, related to the first remark of the referee. We agree that the measurements that were provided in the submitted manuscript were not sufficiently significant in this regard. As said, we made the experiment again, in a different configuration (bandwidth of 3.5 GHz). As shown on fig. 9, no significant change in the height and the width of the correlation peaks could be observed, meaning that the coherence is preserved over durations of 100 μ s (i.e. more than 1100 consecutive waveforms). This property enables pulse integration or accumulation, as rightly indicated by the referee. We mention this possibility in the introduction of the revised manuscript.

4. The authors should check equation (1). To make equation correct, I think the AOFS should be placed before the EDFA, otherwise, another parameter should be defined to represent the time delay from the AOFS to the OI and should be presented in (1).

Eq. 1 has been established using the mathematical framework of a previous paper, entitled "Theory of Talbot lasers" (Guillet de Chatellus, et al., Phys. Rev. A 88, 033828 (2013)). In this case, the AOFS was placed at the input of the loop, and played also the role of the input/output coupler, through leaky diffraction zero orders. In our present manuscript, for simplicity reasons, we kept the mathematical descriptions of the mentioned PRA paper, but slightly changed the experimental set-up, by placing the AOFS inside the loop – not at the input. In fact, we can show mathematically that this difference does not bring any change in the expression of the electric field, except for an unimportant linear phase term, which only translates into a temporal shift of the output signal. So we have preferred to keep the theoretical expression of the electric field as such. For visualization purposes and consistency with the text, we also modified the ordering of the elements in the FSL on Fig. 2. This change re-place the AOFS closer to the input of the FSL, in accordance with the referee's suggestion.

5. The authors should check the experiment parameters. For instance, $\Delta f_s \times \tau_c$ calculated from the experiment parameters does not equal to the results you provided in the paper.

We thank the referee for pointing out this discrepancy. We believe that the reviewer refers to the first lines of section 2B, where multiplying the provided values of Δf_s and τ_c did not lead to the given numerical value. This was actually due to the fact that we did not provide the value of τ_c with a sufficient number of decimals. We corrected this point in the whole manuscript, by providing values of τ_c with 5 significant digits, similarly to the level of precision of f_s . Following these changes, all numerical values in the manuscript have been double checked, and modified accordingly.

Reviewer #2 (Remarks to the Author):

Some technical comments:

Abstract:

- "state-of-the-art photonic techniques ... limited tuning performance". This is not quite accurate. Many existing methods offer reconfigurable/programmable capability for arbitrary waveform generation. Some relevant references should be cited.

Several examples of generation of RFCW using photonic techniques, including those that offer reconfigurable capabilities, are given in the Introduction paragraph, but not in the Abstract. We believe that doing so renders the reading of the Abstract easier.

- "a wide range of real-world applications." This statement is too generic. Some solid examples should be identified.

This statement appears at the end of the Abstract. However due to the length constraints set by the Journal, we could not provide any further details in the Abstract. However, in the Introduction paragraph, we provide concrete examples and references of real-world applications where RFCWs are required, including radar systems, biomedical imaging, testing of RF components and chirped pulse Fourier transform microwave spectroscopy.

Introduction:

- "corresponding to the equivalent of dispersive propagation through thousands of km of conventional single-mode fiber". Dispersive propagation is not clear. Does it particularly refer to the proposed method or existing ones, such as frequency-to-time mapping?

We rewrote the paragraph to make it more clear. The idea is that optical chirped waveforms, such as the ones produced at the output of the FSL, could also be produced by propagation of a phase-locked pulse train through a medium showing group velocity dispersion. However, it turns out that this solution may not be practical, due to the extremely high value of the total group delay dispersion that would be required to match the specifications directly provided by the FSL system. We rewrote the sentence as: "Notice that obtaining equivalent chirp rates by

linear dispersive propagation would require thousands of km of conventional single-mode fiber.”

Theoretical Description:

- The overall shape of optical comb spectrum $g(\omega)$ is an important parameter. How it can be determined and designed?

As written in the text, the envelope of the comb g is set by the gain and losses in the loop. We have chosen not to elaborate further on this issue, since this would require a much more complete theoretical treatment of the system, including a detailed analysis of the properties of the gain medium. We are currently investigating these effects, but in our opinion, they are well beyond the scope of our present paper. Moreover, for practical applications, the shape of g is not so critical, since the envelope of the RF chirps can be shaped arbitrarily by the pulse shaper (PS) at the output of the FSL.

- More information is needed to support the statement of Fourier transform-limited output. Why an integer product of $f_s \tau_c$ guarantees this?

To make things more clear on this topic (section 1A), we added the following sentence:

“Notice that in the case where $\Delta f_s = 0$ (i.e., when the product $f_s \tau_c$ is an integer), all spectral components in Eq. (2) have the same phase, which ensures that optical pulses obtained at the output of the FSL are transform-limited.”

Experimental Results:

- Are the waveforms shown in Fig. 5b,c single-shot measurement or averaged results?

All the waveforms shown in the manuscript are single-shot measurement (no averaging is performed). The following sentence has been added to the caption of Fig. 5: “All traces are single-shot (no averaged) measurements.”

- In Fig. 5d, green curve shows the time-mapped optical power spectrum. Time scale should be given.

Indeed, this is a good point. We added the time scale on Fig. 5d.

- Caption of Fig. 5, the first b should be a.

This typo has been corrected.

- In section D, an optical spectral shaper was used to tailor the temporal envelope. What is the spectral resolution of this shaper?

The spectral resolution of the pulse shaper is 0.1 nm (i.e. 12.5 GHz at 1550 nm). This information has been added in the Methods (#3).

- the acousto-optic frequency shifter is a key component to generate optical frequency combs. Two are used in configuration #2. What is the diffraction efficiency or the overall insertion loss of this device in the loop?

The insertion losses for a single AOFS are about 2 dB, and 4 dB for two AOFS. This information has been added to the Methods sections (#3, #4).

Reviewer #3 (Remarks to the Author):

Some derivations in the paper require a more detailed development. For instance those leading to Eq(3), that leading to the expression of the temporal duration in section B.3, and Eq(13) in Methods.

We thank the referee for bringing up this point. Following his/her advice, we added a paragraph to the Methods (section #1), to recall the expression of a phase-locked pulse train after propagation through a linear dispersive medium. These calculations are derived from Saleh and Teich, Fundamentals of Photonics, (John Wiley & Sons, 2007), Chap. 22.

Regarding the expression of the temporal duration in section B3, we added the following sentence to make things more clear: "As shown in Eq. (7), the temporal envelope of the RF chirps maps the spectral envelope of the FSL comb. Then the temporal duration of the RF chirps τ is approximately equal to the spectral bandwidth of the FSL comb multiplied by the frequency-to-time mapping coefficient (group-velocity dispersion): $\tau = N f_s \times 2\pi |D| = N \left| \frac{\Delta f_s}{f_s} \right| \tau_c$ ".

Finally, to explain how Eq. (13) is obtained (the new reference of this equation is Eq. (18)), we added the following statement before the equation: "If satisfied, the variation of the term $e^{\frac{t\tau^2}{2D^2}}$ in the integral can be neglected, which leads to the following expression:..."

There is also some terminology that should be explained, which might not be straightforward to the non-specialized reader. For example, what is the meaning of the "individual output waveforms"? The authors could try, for example to include an expression of the total field in terms of the sum of individual contributions, to make this picture more clear. regarding terminology, the authors use either ring or loop adjectives. To me a fiber loop is a Sagnac and not a ring cavity.

The expression "individual output waveforms" was used just before Eq. (4). According to the referee's remark, and to avoid any ambiguity, we changed it to: "the individual temporally stretched output waveforms", since the same expression is used 5-6 lines earlier in the text.

According to the pertinent referee's suggestion, we also have derived an expression of the total field in terms of the sum of individual contributions. Indeed, this approach leads to simpler mathematics, and reduces the number of mathematical variables in the manuscript (for instance

it enables to eliminate the electric field and the intensity of individual output waveforms, $E_{\text{ind}}(t)$ and $I_{\text{ind}}(t)$ respectively, that were introduced in the original version of the manuscript).

In this regard, we have organized the first two paragraphs of the Methods section as follows. In the first one (Methods #1), we show that the expression of the electric field at the output of the FSL is similar to that of a phase-locked pulse train after propagation through a dispersive medium exhibiting group-velocity dispersion. More precisely, we derive the expression of the equivalent total group delay dispersion. In Methods #2, we use this analogy to derive the expression of the electric field generated in the FSL, by means of the impulse response of dispersion operator. This leads to an expression of the total field in terms of the sum of individual contributions, as suggested by the referee.

Finally, we did not use the terms “ring”, nor “cavity” in the manuscript. We always refer to the proposed scheme as a “loop”.

All in all, the main problem that I see is that the model includes a considerable number of parameters and it is not easy to see how manipulating one or various of them result in the changing of one or different features of the Chirped RF pulses. I suggest that the authors draft a sort of "design table" where they can put in the first column the relevant design parameters (τ_c , F_s , Δf , etc...) and in the first row the relevant features to be reconfigured (chirp rate, central frequency, pulse bandwidth, pulse shape etc...). This would make things much more clear. On another table they could include some information regarding realistic values of these parameters to achieve certain ranges of given parameters and for which application they might suit. I give an example: a certain range of bandwidth and delays might be useful for instance for SAR Radar, other could be useful for some bio application and so on. These two last suggestions could also illustrate and support the text of a much missed discussion section.

We thank the referee for his point of view. We have tried to provide a description of the properties of the concept as complete as possible, resulting in a relatively dense manuscript. To try to make things more clear and easy to read, as per the suggestions from the referee, we have merged Fig. (3) and Fig. (4). Instead, we have replaced Fig. (3) by a new figure, which essentially provides an illustrated summary of the central design equations that is suggested by the referee (i.e., generated chirp parameters vs. design specifications). In particular, Fig. 3 includes two different sections: the top one is the plot in the time-frequency plane of the optical chirps generated in the FSL. It recalls the mechanism of generation of the RF chirps from the optical ones, by heterodyning with the seed laser. It also shows how the pulse shaper is used to tailor the spectral envelope, the spectral bandwidth and the central frequency of the RF chirps. The second part of the figure recalls the expression of the main parameters of the generated chirp waveform, as a function of the design specifications. We believe that presenting the main results in such a synthetic manner should improve the readability of the manuscript, in line with the comments made by the referee. Finally, following the referee's suggestion, we modified the “Conclusion” section into a “Discussion and conclusion” section. In doing so, we added several points of discussion, including for example, the possibility of reducing, or increasing the period of the RFCW produced in the FSL, to match the needed specifications for a targeted application.

REVIEWERS' COMMENTS:

Reviewer #1 (Remarks to the Author):

The authors answered all my questions. The paper can now be accepted for publication.

Reviewer #2 (Remarks to the Author):

The revised paper has a better shape with most of my questions addressed.

My first comment, as follows, has not been addressed by the authors.

I am convinced that the proposed approach offers essential benefits compared to some existing photonic methods reported in [12-17] for chirped RF waveform generation. Generation of various chirped RF waveforms is import, which has been demonstrated in this work. The capability of arbitrary waveform generation is also highly demanded and has been widely achieved by other approaches. Short of this capability may limit its significance. Comments on this issue will help the readers better understand this method.

In Abstract:

- "state-of-the-art photonic techniques ... limited tuning performance". This is not quite accurate. Many existing methods offer reconfigurable/programmable capability for arbitrary waveform generation. As the authors responded, several examples of generation of RFCW using photonic techniques, including those that offer reconfigurable capabilities, are given in the Introduction paragraph. Therefore, the statement on the existing approaches "as well as limited tuning performance" should be removed.

Reviewer #3 (Remarks to the Author):

I have checked the revisions made by the authors to my previous comments and found them satisfactory. I believe that in its present form the manuscript is much more clear and especially, more easy to follow and understand by the non-specialized reader. I can thus can be recommended for publication.

Reviewer #1 (Remarks to the Author):

The authors answered all my questions. The paper can now be accepted for publication.

Reviewer #2 (Remarks to the Author):

The revised paper has a better shape with most of my questions addressed.

My first comment, as follows, has not been addressed by the authors.

I am convinced that the proposed approach offers essential benefits compared to some existing photonic methods reported in [12-17] for chirped RF waveform generation. Generation of various chirped RF waveforms is important, which has been demonstrated in this work. The capability of arbitrary waveform generation is also highly demanded and has been widely achieved by other approaches. Short of this capability may limit its significance. Comments on this issue will help the readers better understand this method.

This is an interesting point indeed. As such – and as described in this manuscript-, our platform enables the generation of reconfigurable and arbitrary RF chirped waveforms. It turns out that in a slightly different configuration, frequency shifting loops can be a valuable solution for the generation of RF arbitrary signals in a large sense (i.e. not restricted to chirps). This novel feature is currently getting investigated by our research groups, but this capability stays beyond the frame of the manuscript. We believe that the fact that the work reported in the manuscript is restricted to the generation of arbitrary chirps, is made clear enough in the introduction.

In Abstract:

- “state-of-the-art photonic techniques ... limited tuning performance”. This is not quite accurate. Many existing methods offer reconfigurable/programmable capability for arbitrary waveform generation. As the authors responded, several examples of generation of RFCW using photonic techniques, including those that offer reconfigurable capabilities, are given in the Introduction paragraph. Therefore, the statement on the existing approaches “as well as limited tuning performance” should be removed.

We agree that this statement might sound a little bit too negative. According to the reviewer’s suggestion, we have removed it from the abstract.

Reviewer #3 (Remarks to the Author):

I have checked the revisions made by the authors to my previous comments and found them satisfactory. I believe that in its present form the manuscript is much more clear and especially, more easy to follow and understand by the non-specialized reader. I can thus can be recommended for publication.

Sincerely,

Hugues Guillet de Chatellus, on behalf of the authors.